# Direct Enhancement Effect of Hippocampal Cholinergic Neurostimulating Peptide on Cholinergic Activity in the Hippocampus

**DOI:** 10.3390/ijms24108916

**Published:** 2023-05-17

**Authors:** Yuta Madokoro, Daisuke Kato, Yo Tsuda, Itsumi Arakawa, Kengo Suzuki, Toyohiro Sato, Masayuki Mizuno, Yuto Uchida, Kosei Ojika, Noriyuki Matsukawa

**Affiliations:** Department of Neurology, Nagoya City University, 1 Kawasumi, Mizuho-cho, Mizuho-ku, Nagoya 467-8601, Japan

**Keywords:** hippocampal cholinergic neurostimulating peptide, hippocampal cholinergic neurostimulating peptide precursor protein, hippocampal HCNP-pp conditional KO mouse model, choline acetyltransferase, local field potential, theta oscillation, nerve growth factor

## Abstract

The cholinergic efferent network from the medial septal nucleus to the hippocampus is crucial for learning and memory. This study aimed to clarify whether hippocampal cholinergic neurostimulating peptide (HCNP) has a rescue function in the cholinergic dysfunction of HCNP precursor protein (HCNP-pp) conditional knockout (cKO). Chemically synthesized HCNP or a vehicle were continuously administered into the cerebral ventricle of HCNP-pp cKO mice and littermate floxed (control) mice for two weeks via osmotic pumps. We immunohistochemically measured the cholinergic axon volume in the stratum oriens and functionally evaluated the local field potential in the CA1. Furthermore, choline acetyltransferase (ChAT) and nerve growth factor (NGF) receptor (TrkA and p75NTR) abundances were quantified in wild-type (WT) mice administered HCNP or the vehicle. As a result, HCNP administration morphologically increased the cholinergic axonal volume and electrophysiological theta power in HCNP-pp cKO and control mice. Following the administration of HCNP to WT mice, TrkA and p75NTR levels also decreased significantly. These data suggest that extrinsic HCNP may compensate for the reduced cholinergic axonal volume and theta power in HCNP-pp cKO mice. HCNP may function complementarily to NGF in the cholinergic network in vivo. HCNP may represent a therapeutic candidate for neurological diseases with cholinergic dysfunction, e.g., Alzheimer’s disease and Lewy body dementia.

## 1. Introduction

Hippocampal glutamatergic neuronal activity plays a crucial role in episodic and learning memory [1]. Hippocampal function is modified via the communication of afferent and efferent neural networks from associated areas [2]. The timing-dependent synchronized neural firing of multiple efferent fibers as a group in the hippocampus is involved in the encoding of memory [3]. In this way, the local field potential (LFP) can be generated by summating the electrical activity from a number of neurons in the local field [4]. Theta oscillation (4–12 Hz) and gamma oscillation (25–100 Hz), as synchronized rhythmic activities, are inextricably involved in hippocampal function [4]. In particular, the theta power in the hippocampus plays a crucial role in working memory, decision making, and memory consolidation [5,6,7]. Theta oscillation is created by the interaction of the GABAergic and cholinergic projection from the medial septal nucleus (MSN) to the hippocampus [8,9,10,11].

Hippocampal cholinergic neurostimulating peptide (HCNP), which induces acetylcholine (Ach) synthesis in MSN, was originally isolated and purified from the soluble fraction of the hippocampus of juvenile–adult rats [12]. HCNP is formed at the N-terminal region of the 21-kD HCNP precursor protein (HCNP-pp) and comprises 186 amino acids [13]. Recently, we reported that HCNP-pp conditional knockout (HCNP-pp cKO) mice exhibit reduced theta activity via a decrease in choline acetyltransferase (ChAT)-positive axonal volume in the stratum oriens of the hippocampal CA1 [14], as well as ra educed Ach concentration in the hippocampus [15]. The dysfunction of glutamatergic neural activation was consequently observed in the hippocampus of HCNP-pp cKO mice [16]. Hence, HCNP might serve as a regulator of hippocampal cholinergic function. Moreover, HCNP-pp cKO mice represent an appropriate genetic model for cholinergic functional impairment in septo-hippocampal interactions. When considering the development of a novel therapy using HCNP against septo-hippocampal cholinergic dysfunction, the effect of extrinsic HCNP administration on cholinergic dysfunction should be tested in this genetic model.

Accordingly, the current study seeks to test the involvement of HCNP reduction in cholinergic dysfunction in HCNP-pp cKO mice and to verify its potential as a therapeutic candidate for diseases with cholinergic impairment, including Alzheimer’s disease (AD) and Lewy body dementia.

## 2. Results

### 2.1. HCNP Administration Recovers ChAT-Positive Axons in HCNP-pp cKO Mice

Based on a semi-quantitative analysis of ChAT-positive axonal volume using IMARIS 9.2.0 software, we previously reported that cholinergic projections from the MSN to the hippocampus were regionally diminished in the stratum oriens of HCNP-pp cKO mice [14]. Therefore, in the current study, to assess whether extrinsic HCNP administration could counteract cholinergic impairment in the hippocampi of HCNP-pp cKO mice, we continuously administered HCNP or vehicle–PBS into the ventricles of HCNP-pp cKO mice or HCNP-pp floxed (Control) mice for 2 weeks. Following the treatment period, the ChAT-positive axonal volume in the stratum oriens was significantly decreased in the vehicle-treated KO mice (KO-Vehicle) compared to the vehicle-treated control mice (Control-Vehicle; Figure 1a,b, Control-Vehicle: 1296.2 ± 43.6 μm^3^, *n* = 36 vs. KO-Vehicle: 1014.3 ± 55.7 μm^3^, *n* = 48, *p* < 0.001, two-way ANOVA with the Holm post-hoc test), which is consistent with our previous report [14]. In contrast, the ChAT-positive axonal volume in the stratum oriens of the HCNP-treated KO mice (KO-HCNP) significantly recovered to a level similar to the level found in the Control-Vehicle mice (Figure 1a,b, Control-Vehicle: 1296.2 ± 43.6 μm^3^, *n* = 36 vs. KO-HCNP: 1193.6 ± 44.6 μm^3^, *n* = 48, *p* = 0.155). A similar enhancement effect for ChAT-positive axons was observed in the stratum oriens of the HCNP-treated control mice (Control-HCNP) compared to the Control-Vehicle group (Figure 1a,b, Control-Vehicle: 1296.2 ± 43.6 μm^3^, *n* = 36 vs. Control-HCNP: 1505.0 ± 46.1 μm^3^, *n* = 47, *p* < 0.05).

To further assess the effect of HCNP administration on cholinergic axon volume, we analyzed the length, cross-sectional area, and number of branching points of the cholinergic axons. Although there was no significant change in the cross-sectional area of the cholinergic axon terminal, HCNP administration significantly increased the length (Control-Vehicle: 1774.2 ± 56.4 μm, *n* = 36 vs. Control-HCNP: 1998.1 ± 60.5 μm, *n* = 47, *p* < 0.05, KO-Vehicle: 1313.7 ± 63.1 μm, *n* = 48, vs. KO-HCNP: 1579.3 ± 56.0 μm, *p* < 0.01) and the number of branching points (Control-Vehicle: 163.1 ± 6.9, *n* = 36 vs. Control-HCNP: 200.7 ± 7.9, *n* = 47, *p* < 0.05, KO-Vehicle: 113.6 ± 7.6, *n* = 48, vs. KO-HCNP: 141.9 ± 6.9, *p* < 0.05) in the Control-HCNP and KO-HCNP groups compared to the Control-Vehicle and KO-Vehicle groups, respectively (Figure 1c).

### 2.2. HCNP Administration Recovers Hippocampal Theta Power

We previously reported that theta oscillations related to septo-hippocampal cholinergic function were significantly inhibited in the hippocampus of HCNP-pp cKO mice [3]. Therefore, we herein assessed theta power in a functional cholinergic assay as a recovery effect of HCNP administration. Two weeks after the administration of the vehicle or HCNP, we carefully inserted the silicon probes stereotactically into the CA1 pyramidal layer, recorded the LFP for 3 min, and analyzed the theta power (total theta: 3–12 Hz). The KO-Vehicle mice showed a significantly lower total theta power than the Control-Vehicle mice (Figure 2a,b, Control-Vehicle: 9.44 ± 0.99 × 10^−4^ mV^2^/Hz, *n* = 20 trials vs. KO-Vehicle: 6.26 ± 0.78 × 10^−4^ mV^2^/Hz, *n* = 20 trials, *p* < 0.05, two-way ANOVA with the Holm post-hoc test), which was consistent with our previous work [14]. The Control-HCNP mice showed a significantly higher total theta power than the Control-Vehicle mice (Figure 2a,b, Control-Vehicle: 9.44 ± 0.99 × 10^−4^ mV^2^/Hz, *n* = 20 trials vs. Control-HCNP: 13.4 ± 0.54 × 10^−4^ mV^2^/Hz, *n* = 17 trials, *p* < 0.01). Unexpectedly, the KO-HCNP mice did not have a higher total theta power than the KO-Vehicle mice. However, HCNP administration significantly enhanced the type 2 theta (theta2; Figure 2c, Control-Vehicle: 8.74 ± 1.37 × 10^−4^ mV^2^/Hz, *n* = 20 trials vs. Control-HCNP: 20.7 ± 1.15 × 10^−4^ mV^2^/Hz, *n* = 17 trials, *p* < 0.001; KO-Vehicle: 4.59 ± 0.65 × 10^−4^ mV^2^/Hz, *n* = 20 trials vs. KO-HCNP: 9.07 ± 0.16 × 10^−4^ mV^2^/Hz, *n* = 20 trials, *p* < 0.05)—a subtype of theta ranging from 4 to 9 Hz which more strongly reflects cholinergic activity [17]—in the control and KO mice. Notably, the KO-HCNP mice exhibited a relatively similar level of theta2 power to that of the Control-Vehicle mice (Figure 2c).

### 2.3. HCNP Administration Decreases the Abundance of Nerve Growth Factor (NGF) Receptors, TrkA, and p75NTR

Many researchers have reported that NGF is involved in the maintenance and preservation of cholinergic axonal terminals [18,19,20,21,22,23,24]. We also reported that HCNP and NGF might have cooperative roles in the biochemical differentiation of cholinergic neurons in the MSN [25]. Therefore, to screen the relative function of NGF in the observed enhancement effect, we evaluated the levels of ChAT and NGF receptors, TrkA, and p75NTR, following HCNP or vehicle administration to wild-type (WT) mice. The administration of HCNP tended to increase the ChAT level; however, the increase was not significant (WT-Vehicle: 0.396 ± 0.014, *n* = 5 vs. WT-HCNP: 0.541 ± 0.064, *n* = 5, *p* = 0.0584, Student’s *t*-test). In contrast, TrkA (WT-Vehicle: 1.27 ± 0.056, *n* = 5 vs. WT-HCNP: 0.721 ± 0.076, *n* = 5, *p* < 0.01) and p75NTR (WT-Vehicle: 0.969 ± 0.060, *n* = 5 vs. WT-HCNP: 0.675 ± 0.074, *n* = 5, *p* < 0.05) levels were significantly lower in the WT-HCNP mice than in the WT-Vehicle mice (Figure 3).

## 3. Discussion

This study confirmed three new findings. First, the direct administration of HCNP into the cerebral ventricles might increase the volume of ChAT-positive axonal terminals in the stratum oriens of CA1 in HCNP-pp cKO and Control mice. Second, type 2 theta is also induced by HCNP administration in the hippocampus of HCNP-pp cKO and Control mice. Third, HCNP administration into the ventricles may significantly reduce the amounts of both NGF-related receptors, TrkA and p75NTR, whereas ChAT tends to increase them, although not significantly.

Cholinergic neurons in the basal forebrain, including the MSN, undergo selective degeneration and gradually disappear in the early stages of AD [26,27,28,29,30,31], Parkinson’s disease with dementia (PDD) [27,32,33,34], and dementia with Lewy bodies (DLB) [27,34,35]. Meanwhile, cholinesterase inhibitors (ChE-Is) clinically ameliorate the pathological symptoms in patients with AD or DLB [27,34,35]. However, the effect of ChE-Is is short-lived as the enhancing effect of ChE-Is on Ach within neuronal clefts is dependent on Ach production in cholinergic neuronal terminals. Hence, the gradual neurodegeneration of cholinergic neurons in the MSN may reduce the therapeutic response to ChE-Is. Therefore, research efforts have been made to maintain cholinergic neurons from the point of view of neurotrophic factors, such as NGF [19], brain-derived neurotrophic factor [36], and bone morphogenetic protein-9 [37]. In the current study, the administration of the functional peptide, HCNP, was found to neurophysiologically counteract cholinergic impairment by inducing a regional increase in ChAT-positive neuronal terminals within the hippocampi of HCNP-pp cKO mice. However, the trophic effect of HCNP has not yet been directly confirmed in an animal model [15]. In fact, HCNP was initially isolated from embryonic day 14 rat hippocampi [12] as a peptide for enhancing neurite outgrowth [13]. The increase in cholinergic axonal length and branching points but not thickness observed in this experiment is likely consistent with previous work [13]. In particular, the increase in theta 2 power in the hippocampus with cholinergic projection may suggest that the administration of HCNP morphologically and electrophysiologically enhances the specific cholinergic function in the septo-hippocampal interaction [17]. Notably, these cholinergic parameters in the KO-HCNP mice improved to a similar extent as in the Control-Vehicle mice. Thus, these data, in combination with our previously reported findings, further support the hypothesis that HCNP serves as a functional regulator of neural cholinergic activity in septo-hippocampal interactions.

We previously reported that HCNP might induce acetylcholine synthesis complementarily with NGF in WT explant culture tissue of the MSN [25]. Considering that NGF is a typical neurotrophic factor for cholinergic neurons in the hippocampus [18,19,20,21,22,23,24,38], we investigated NGF receptor expression (TrkA and p75NTR) to elucidate the direct involvement of HCNP administration in NGF receptor expression. NGF and NGF receptors have also been reported to change owing to a compensatory reaction against cholinergic dysfunction [39]. Considering that the expression of NGF receptors might primarily change in the hippocampus of HCNP-pp cKO mice, we used WT mice, not HCNP-pp cKO mice, to assess the pure effect of HCNP administration in this study. Interestingly, both NGF-related receptors were significantly decreased in the mice that received HCNP, suggesting that the increased ChAT in this model was induced independently of NGF signals. These findings support our previous report that HCNP enhanced cholinergic function by regulating ChAT synthesis with NGF in an explant culture using rat MS [25]. Accordingly, the HCNP-induced reduction in NGF receptor expression may result from negative feedback against excessively increased cholinergic function, as HCNP and NGF can function in cholinergic neurons in parallel.

When considering the development of a novel therapy using HCNP against septo-hippocampal cholinergic dysfunction in humans, it is necessary to design an appropriate method for effectively administrating the 11-amino-acid peptide without requiring direct injection into the cerebral ventricles. One potential option is a transnasal formula [40] or the oral administration of the HCNP agonist. Nevertheless, to generate HCNP agonists, it is necessary to first characterize and isolate HCNP receptors, which has not yet been reported; however, it might exist in the crude P2 membrane fraction [41]. NGF or its metabolic pathway have also been proposed as potential targets for arresting the degeneration of the cholinergic system by boosting trophic influence; however, the intracerebral and exogenous application of NGF has not yet been proven successful [38].

Certain limitations were noted in this study. First, we performed all experiments with old mice. Thus, if the phenotype changes with age in the HCNP-pp cKO mice, as in our previous report [16], different effects may be generated following HCNP administration in younger mice. Second, the LFP was measured one day after suspending HCNP administration. We could not rule out the possibility that removing the osmotic pump may have affected the baseline brain activity, including the total theta power. Third, the possibility that urethane anesthesia affected theta oscillation cannot be excluded. Urethane anesthesia may diminish type 1 theta, mainly through the inhibition of glutamatergic neuronal activity [42], activating type-2-theta-related cholinergic neuronal activity. As type 2 theta is inhibited by atropine or scopolamine [43,44], it reflects cholinergic activity and is widely used as an indicator of cholinergic activity [10,45,46]. We must constantly consider any influence of urethane anesthesia on type 2 theta via any neurotransmitter, including nicotinic receptors. Fourth, the amount of NGF was not measured in this study. We could not show the involvement of NGF alteration in the downregulation of NGF receptors by HCNP administration. The possibility that HCNP directly affected NGF receptors cannot be ruled out. Fifth, we confirmed the enhancement effect of HCNP administration on cholinergic neurons in the same strains of HCNP-pp cKO mice compared to the control-HCNP-pp littermate floxed mice. Meanwhile, WT mice were used in the NGF-related receptor study; hence, the change in the number of NGF receptors might be derived from differences between the WT and HCNP-pp littermate floxed mice. To investigate the potential of HCNP as a therapeutic target for dementia, additional studies are required, including the assessment of NGF receptor kinetics in HCNP-pp cKO mice, to better characterize the features of HCNP.

## 4. Materials and Methods

### 4.1. Animals

In this study, we used the same strain of HCNP-pp cKO mice as described previously [14]. Briefly, HCNP-pp knockout was achieved following the Cre-loxP system with the Cre-ERT gene, which was excised between the middle portion of exon 1 and the intron that lies between exons 3 and 4via Cre recombinase. Tamoxifen was injected three months after birth into floxed HCNP-pp mice (CreERT/+, fHCNP-pp+/+), called HCNP-pp cKO mice, and littermate control mice (CreERT/, fHCNP+/+ or/+). We used 24 male mice for LFP measurement and the immunohistochemical analyses; data were obtained from 15 mice (87–91 weeks old) divided into four groups: Control-Vehicle (*n* = 4), Control-HCNP (*n* = 3), KO-Vehicle (*n* = 4), and KO-HCNP (*n* = 4). Additionally, ten WT (C57BL/6) male mice (23 weeks old; WT-Vehicle (*n* = 5), WT-HCNP (*n* = 5)) were used for the Western blot experiments. The animals were housed under specific-pathogen-free conditions with a 12 h light/dark cycle (with the lights on from 08:00 to 20:00) and provided with free access to food and water.

### 4.2. HCNP Administration

We chemically synthesized human HCNP (PVDLSKWSGPL). Under anesthesia with 1% isoflurane, a midsagittal incision was made on the scalp, and a subcutaneous tunnel was opened between the shoulder blades, where the Alzet osmotic pumps (model 1002; pumping rate 0.25 μL/h, DURECT Corporation, Cupertino, CA, USA) were implanted. Subsequently, a craniotomy at 0.6 mm posterior and 1.2 mm lateral from the bregma was performed stereotactically. The tip of the pump was carefully inserted into the craniotomy site and was angled at 8°. The cannula was fixed in place using dental cement. The incision was closed with silk sutures and dabbed with Vetbond (3M, St Paul, MN, USA). Intraventricular infusion of the vehicle (bicarbonate buffer) or 0.75 pg/h of HCNP was then performed for 14 d.

### 4.3. Quantification of Hippocampal Cholinergic Axons

The protocol for quantifying hippocampal cholinergic axons was performed in accordance with our previous report [14]. Briefly, after fixation in 4% paraformaldehyde/phosphate buffer (PB, pH 7.4), the mouse brains were equilibrated in a 30% sucrose solution/PB and sectioned at thicknesses of 20 μm using a cryostat (Leica Microsystems, Bensheim, Germany).

We calculated the cholinergic axonal volume in 20 μm thick coronal brain sections at −2.3 mm from the bregma, focusing on the stratum oriens of CA1. Before performing ChAT immunofluorescence, each section was treated with the TrueBlack Lipofuscin Autofluorescence Quencher (Biotium Inc., Fremont, CA, USA). The sections were incubated for 24 h with a primary goat anti-ChAT (polyclonal) antibody (catalog number AB144P, 1:100; Merck-Millipore, Billerica, MA, USA) at 20 °C. The sections were then incubated with a secondary Alexa Fluor-594 donkey anti-goat IgG antibody (1:500; Thermo Fisher Scientific, Waltham, MA, USA) for 1 h at 20 °C. Fluorescence imaging of cholinergic fibers was carried out using an A1Rsi laser confocal microscope (Nikon, Tokyo, Japan). Eight-micron-thick Z-stacks were acquired at 0.4 μm intervals for each section within the stratum oriens layer of the CA1 field. Two sections of each mouse hippocampal slice were prepared at a given location, while three regions of interest were selected per side in the stratum oriens. The images were transferred to IMARIS 9.2.0 (Bitplane, Zurich, Switzerland), and the ChAT-positive axonal volume, length, cross-sectional area, and the number of branch points were calculated using an empirically optimized batch protocol. IMARIS enables the 3D reconstruction of neurons and arborization analysis. It is a technique that enables visualizing various structures, such as axons and dendrites, somas, and dendritic spines. Further, IMARIS automatically calculates a range of neuron-specific measurements, such as the volume of specific structures, dendrite or segment length, branch level, diameter, and cross-sectional area (https://imaris.oxinst.com/products/imaris-for-neuroscientists accessed on 28 February 2023). The total axonal volume (μm^3^), length (μm), cross-sectional area (μm^2^), and the number of branch points were summed per image/subject and then exported to an Excel spreadsheet.

### 4.4. Electrophysiological Recordings

The technique used in this experiment was described in our previous report [14]. Briefly, all mice were anesthetized via an intraperitoneal injection of ketamine (74 mg/kg) and xylazine (10 mg/kg). During surgery, the mice were placed on a heating pad, and eye ointment (Tarivid ophthalmic ointment 0.3%; Santen Pharmaceutical Co., Ltd., Osaka, Japan) was used to prevent corneal drying. After removing the osmotic pump for the administration of HCNP or the vehicle, a head plate was firmly attached to the dental cement (Fuji lute BC; GC, Tokyo, Japan, Bistite II; Tokuyama Dental, Tokyo, Japan). The mice were allowed to recover for one day before LFP recording.

For the LFP experiments, dexamethasone sodium phosphate (1.32 mg/kg) was administered intraperitoneally 1 h before surgery to prevent cerebral edema. Under anesthesia with 1.6 g/kg urethane, a ~2 mm diameter circular craniotomy (AP: −2.0 mm, ML: −1.3 mm) was performed. To prevent the exposed brain surface from drying and to reduce noise during recording, we applied artificial CSF (NaCl, 125 mM; KCl, 3.5 mM; NaH_2_PO_4_, 1.25 mM; NaHCO_3_, 26 mM; CaCl_2_, 2 mM; MgCl_2_, 2 mM; D-glucose, 15 mM) to the surface of the exposed brain. A 16-channel silicon probe (A1-16-25-177, NeuroNexus Technologies, Ann Arbor, MI, USA) was stereotactically inserted and advanced stepwise to the target position using a microcontroller. The neural activity of the pyramidal layer in the left CA1 was recorded at depths of 1100–1500 μm from the pia. The stereotactic coordinates were determined by referring to the Paxinos and Franklin atlas (2008) [47]. All in vivo LFP recordings were digitally downsampled to 1 kHz using the Omniplex system (Plexon, Dallas, TX, USA) and filtered at a bandpass of 0.05–200 Hz. Proximity to the hippocampal pyramidal cell layer was determined by (1) the depth of the probe, (2) the presence of action potential discharge, and (3) phase reversal of the LFP at theta frequencies above and below the recording site [48]. Each LFP, which was presumed to be in the pyramidal layer, was recorded for 3 min. The recorded LFP data were extracted using an offline sorter (Plexon), and the power spectral density (PSD) of each LFP dataset was analyzed using NeuroExplorer (Plexon). The average PSD of the LFPs recorded at different electrodes at the same time was analyzed to calculate the PSD of the theta range (theta power). The theta (total theta) and theta2 ranges were defined as 3–12 Hz [49] and 4–9 Hz [17], respectively.

### 4.5. Western Blot Analyses

Western blotting was performed as previously described [14]. Briefly, under deep pentobarbital anesthesia, each mouse was transcardially perfused with PBS. After their brains were removed and placed on ice, the bilateral hippocampi were dissected, immediately frozen in liquid nitrogen, and stored at −80 °C until use. Frozen hippocampi from each of the five WT-Vehicle mice and five WT-HCNP mice were homogenized in a lysis buffer. The homogenates were centrifuged at 15,000× *g* for 3 min at 4 °C. After measuring the protein content using the Bradford assay (Pierce, Rockford, IL, USA), 10 µg of each supernatant fraction was loaded onto each lane of a 10% SDS-PAGE gel. After electrophoresis, the samples were transferred to Hybond-P membranes (GE Healthcare, Tokyo, Japan) and incubated with 1:500 goat polyclonal anti-ChAT antibody (catalog number AB144P; Merck-Millipore, Billerica, MA, USA), 1:1000 rabbit polyclonal anti-TrkA antibody (catalog number 2505; Cell Signaling Technology, Danvers, MA, USA), 1:4000 polyclonal rabbit anti-p75NTR antibody (catalog number 55014-1-AP; Proteintech, Chicago, IL, USA), or 1:100,000 mouse monoclonal anti-β-actin antibody (catalog number A5441; Sigma-Aldrich, St. Louis, MO, USA). The membranes were then probed with horseradish peroxidase-conjugated anti-goat, anti-rabbit, or anti-mouse IgG antibodies. Immunoreactive bands were visualized using the ECL Prime Western Blotting Detection kit (GE Healthcare, Tokyo, Japan) and recorded using an ImageQuant LAS 4000 (GE Healthcare, Tokyo, Japan). Western blots were quantified using the Amersham Imager 600 Analysis Software (GE Healthcare, Tokyo, Japan).

### 4.6. Data Analysis

Data, presented as the mean ± SEM, were analyzed using a two-way ANOVA with the Holm post-hoc test except for Figure 3 (in which Student’s *t*-test was used) to analyze differences between groups.

## 5. Conclusions

Extrinsic HCNP administration into the cerebral ventricles can morphologically increase cholinergic projections into the hippocampus, in addition to the electrophysiological-function-ameliorating functions of the cholinergic network in HCNP-pp cKO mice. HCNP may function complementarily to NGF in vivo in the cholinergic network. Thus, HCNP could be a potential therapeutic candidate for neurological diseases with cholinergic dysfunction, such as AD and DLB.

## Figures and Tables

**Figure 1 ijms-24-08916-f001:**
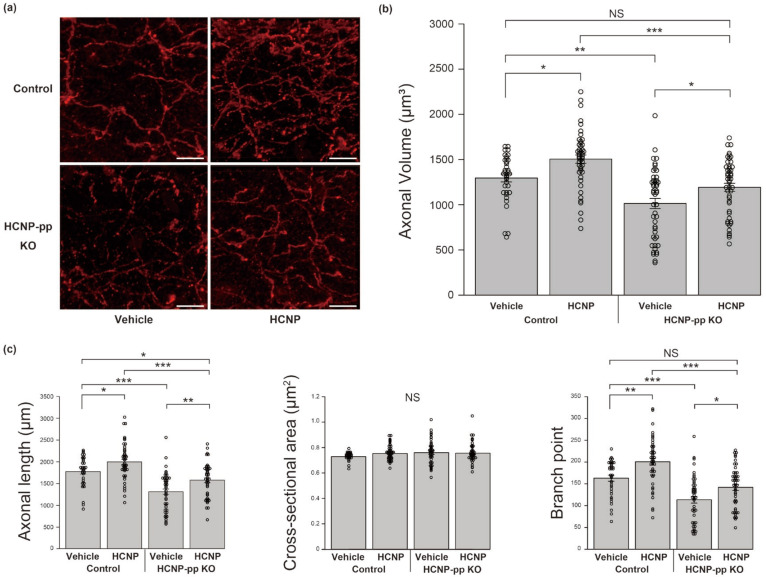
Quantification of ChAT-positive CA1 axons. (**a**) Representative images of ChAT immunohistochemistry in the stratum oriens of control (**upper**) and HCNP-pp cKO (**lower**) mice following administration of vehicle (**left**) and HCNP (**right**). (**b**) Statistical analysis of ChAT-positive axonal volume in the stratum oriens. (**c**) Statistical analyses of the length, cross-sectional area, and the number of branching points of the cholinergic axons in the stratum oriens. Control-Vehicle: *n* = 36, KO-Vehicle: *n* = 48, KO-HCNP: *n* = 48, Control-HCNP: *n* = 47. Data are presented as mean ± standard error of the mean (SEM); two-way ANOVA with Holm post-hoc test. NS = not significant, * *p* < 0.05, ** *p* < 0.01, and *** *p* < 0.001. Scale bar = 10 μm.

**Figure 2 ijms-24-08916-f002:**
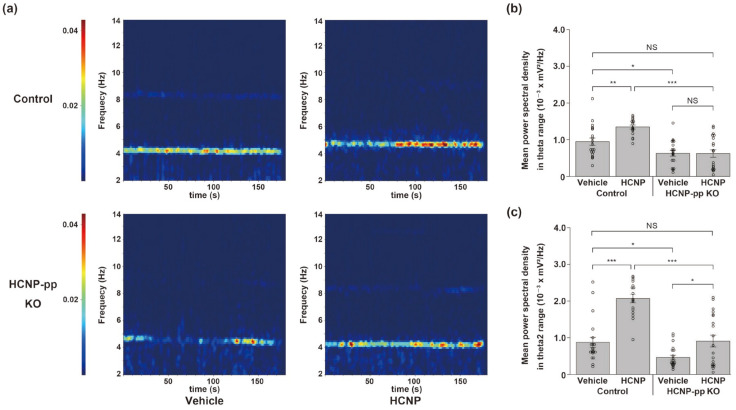
Analysis of local field potential (LFP) in the hippocampus. (**a**) Representative power spectrum of LFP in Control (**upper**) and HCNP-pp cKO (**lower**) following administration of vehicle (**left**) and HCNP (**right**). (**b**) The power spectral density (PSD) of LFP in the CA1 pyramidal layer. The theta range was defined as 3–12 Hz (total theta). (**c**) The PSD of LFP in the CA1 pyramidal layer. The type 2 theta (theta2) range was defined as 4–9 Hz. Control-Vehicle: *n* = 20 trials, KO-Vehicle: *n* = 20 trials, Control-HCNP: *n* = 17 trials, KO-HCNP: *n* = 20 trials. Data are presented as the mean ± SEM; two-way ANOVA with Holm post-hoc test. NS = not significant, * *p* < 0.05, ** *p* < 0.01, and *** *p* < 0.001.

**Figure 3 ijms-24-08916-f003:**
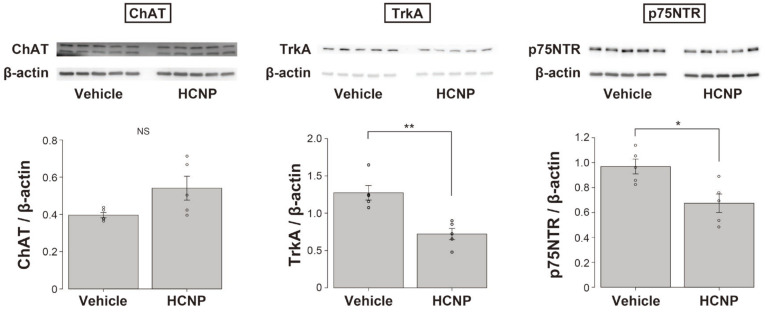
Evaluation of ChAT (**left**), TrkA (**middle**), and p75NTR (**right**) levels following HCNP or vehicle administration in wild-type (WT) mice. WT-Vehicle: *n* = 5; WT-HCNP: *n* = 5. Data are presented as mean ± SEM; Student’s *t*-test. NS = not significant, * *p* < 0.05, ** *p* < 0.01.

## Data Availability

The data presented in this study are available upon request from the corresponding author.

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
