# Peer review of "Direct Enhancement Effect of Hippocampal Cholinergic Neurostimulating Peptide on Cholinergic Activity in the Hippocampus"

_ijms, 2023, doi:10.3390/ijms24108916_

Round 1

Reviewer 1 Report

The manuscript by Madokoro and coworkers describes effects of deletion and substitution, respectively, of the protein HCPN on several structural, molecular and functional properties of the mouse hippocampus. HCPN (hippocampal cholinergic neurostimulating peptide) had previously been shown to enhance cholinergic fiber density/volume and cholinergic signaling in the hippocampus. The authors now ask whether a substitution of HCPN in conditional (?) HCPN-precursor protein-knockout mice can rescue the previously described effects. They find an increase in the axonal volume, length and branching points of cholinergic fibers in stratum oriens, increased oscillation power in a sub-band of theta oscillations, and changes in the protein level of ChAT (upregulation) and NGF-receptors (downregulation). The latter experiment has only been done in WT mice, while the axonal morphology and the theta oscillations have been measured in WT and KO mice, each without and with HCPN administration.

Overall, this appears to be a solid study, based on interesting previous work by the same group and others. The topic is potentially relevant for new treatment strategies in neurodegenerative disorders, as pointed out by the authors. I two comments on the experimental design and several smaller (partly editorial) issues.

 1. Assessment of protein levels: While morphometry and electrophysiology have been conducted symmetrically in WT and KO mice, the proteins have only been studied in WT animals. The reasons given in the third paragraph of the discussion are, in my eyes, not convincing. The authors say that both, NGF and HCNP affect cholinergic innervation, and that HCNP knockout can alone affect the dynamics of NGF receptors. A similar argument does also apply to all other parameters measured. Also, as the authors motivate their study with the potential clinical use of HCNP, they should report its effects in as many situations as possible. Thus, it would be very valuable to test HCNP-effects in the three proteins also in KO mice.

2. Electrophysiological recordings: Field potentials measurements were done after removing the HCNP administration for one day. This may have had severe consequences on baseline activity values in both, WT and KO mice. The lack of effect in total theta power in KO mice may be, in part, due to this situation. While it may be difficult to do the experiment in the continuous presence of the minipump, it might help to perform a control experiment where theta is measured at different times after withdrawal of the pump, beginning as early as possible. This could serve to judge the validity of data presented in Fig. 2. By the way: I was lacking a calibration of the color code in Fig. 2a.

3. I was surprised by the very narrow theta band shown in the time-frequency plots in Fig. 2. In order to get a realistic impression, it would be important to show power distributions over longer stretches. Is it possible that the urethane anesthesia has reduced theta variance?

4. Methods: The method for measuring axonal volume and axonal cross-sectional area has not been explicitly clarified. Since these measures are not generally established, they should be explained.

Specific and minor comments (not ordered by priority)

5. Abstract, line 23: a non-significant ‘result’ should, in my view, not be reported in the paper’s summary.

 6. Abstract, second-to-last sentence: I could not understand the phrase ‘…which are not derived from the NGF system’.

7. Introduction, last 5 lines: At two instances, the authors state that they want to ‘confirm’ certain findings. It might be more neutral to say that statements should be ‘tested’.

8. Reference [8] is a very old reference to theta oscillations, and as such it is very meritful. However, some more recent papers have illustrated the importance of cholinergic septo-hippocampal projections more directly. These may by quoted in addition.

 9. What is counted as ‘n’ in figure 1b/c?

Reviewer 2 Report

In the the current study the auhors seek to confirm the involvement of HCNP reduction  in cholinergic dysfunction in HCNP-pp KO mice and to verify its potential as a therapeutic candidate for diseases with cholinergic impairment, including Alzheimer’s disease 62 (AD) and Lewy body dementia.

The authors confirm the following:

-Direct administration of HCNP into the cerebral ventricles might increase the volume of ChAT-positive axonal terminals in the stratum oriens of CA1 in HCNP-pp KO and Control mice. 

-Type 2 theta is also induced by HCNP administration in the hippocampus of HCNP-pp KO and Control mice.

-HCNP administration into the ventricles may significantly reduce the amounts of both NGF-related receptors, TrkA and p75NTR, whereas ChAT tends to increase, however not significantly.

The manuscript reads well. To raise the level of the manuscript need to discuss the effect on the NGF receptors. Why they are reduced? Is there an increase or decrease of NGF? Does HCNP has a direct effect on the the NGF receptors like binding to the receptors.

Did the authors tried other growth factors like BDNF, IGF-1 which might have more relevance to the cholinergic neurons?

-Dysfunction of glutamatergic neural activation was consequently observed in the hippocampus of HCNP-pp-KO mice. So what happens after injection of HCNP to the glutamate  system?

There are minor spelling mistakes.                                                                                                     
